# Characteristics of Mild Cognitive Impairment and Associated Factors in MSA Patients

**DOI:** 10.3390/brainsci13040582

**Published:** 2023-03-30

**Authors:** Zhihong Shi, Jinhong Zhang, Ping Zhao, Xiyu Li, Shuai Liu, Hao Wu, Peifei Jia, Yong Ji

**Affiliations:** 1Department of Neurology and Tianjin Key Laboratory of Cerebrovascular Disease and Neurodegenerative Disease, Tianjin Dementia Institute, Tianjin Huanhu Hospital, Tianjin University, Tianjin 300350, China; 2Clinical College of Neurology, Neurosurgery and Neurorehabilitation, Tianjin Medical University, Tianjin 300070, China; 3Department of Neurology, Second Hospital of Tianjin Medical University, Tianjin 300211, China; 4China National Clinical Research Center for Neurological Diseases, Beijing 100070, China; 5Department of Neurology, Beijing Tiantan Hospital, Capital Medical University, Beijing 100070, China

**Keywords:** multiple system atrophy, mild cognitive impairment, phenotypes, motor onset

## Abstract

Mild cognitive impairment (MCI) in multiple-system atrophy (MSA) patients is common but remains poorly characterized, and the related factors are unclear. This retrospective study included 200 consecutive patients with a clinical diagnosis of possible or probable MSA, 102 MSA patients with MCI (MSA-MCI), and 98 MSA patients with normal cognition (MSA-NC). Cognitive profiles were compared between MSA-MCI and MSA-NC patients using the MoCA. In addition, demographic as well as major motor and nonmotor symptom differences were compared between MSA-MCI and MSA-NC patients. The median MMSE score was 26 points. Overall, MSA-MCI was observed in 51% of patients, with predominant impairment in visuospatial, executive, and attention functions compared with MSA-NC patients. MSA-MCI patients were older (*p* = 0.015) and had a later onset age (*p* = 0.024) and a higher frequency of hypertension, motor onset, and MSA with the predominant parkinsonism (MSA-P) phenotype than MSA-NC patients. The positive rate of orthostatic hypotension (OH) in MSA-MCI patients was significantly decreased and depression/anxiety was significantly increased compared with MSA-NC patients (*p* = 0.004). Multivariate logistic analysis showed that motor onset was independently associated with MCI in MSA patients. MSA-MCI patients had impairment in visuospatial, executive, and attention functions. More prominent memory impairment was observed in MSA-P than in MSA-C patients. Motor onset was independently associated with MCI in MSA patients. MCI was commonly presented in MSA with more prominent memory impairment in MSA-P. Future follow-up studies are warranted to identify more factors that influence cognitive impairment in MSA.

## 1. Introduction

Multiple system atrophy (MSA) is a sporadic adult-onset neurodegenerative disorder. The main clinical characteristics are parkinsonism, cerebellar ataxia, autonomic failure, and corticospinal impairments [1,2]. Increasing evidence suggests mild cognitive impairment (MCI) is common in MSA. Different from dementia with Lewy bodies, dementia is not a key character. Previous reports showed that cognitive impairment occurs in about 17–47% of MSA patients [3]. In contrast, severe cognitive decline that significantly disrupts daily living is uncommon in MSA, and in autopsy-confirmed MSA it was only 0.5% [1]. In the current diagnostic criteria, dementia within 3 years of disease onset is regarded as a non-supporting feature [4]. It is very important to complete global cognitive tests to clarify the characteristics of MCI in relatively early-stage MSA.

Most studies showed cognitive impairment in MSA patients affects memory, executive, attention, and visuospatial functions, where multiple domain impairment was common in MSA-P [5,6,7]. Comparative studies regarding cognitive impairment in MSA-P and MSA-C subtypes have been conducted with controversial results reported [5,6,7,8,9]. Executive dysfunction was the most prominent cognitive impairment in MSA, especially in MSA-P. The subcortical pattern was similar to Parkinson’s disease (PD). Impaired spontaneous immediate verbal recall that improves with cueing is also a similar feature of MSA-P and PD. However, this feature was not found in MSA-C. Impaired learning is the most prominent memory dysfunction in MSA-C, visuospatial functions are also more prominent in MSA-C than MSA-P. More comparative studies with relatively large samples are needed to investigate the difference in cognitive impairment between both motor subtypes.

The correlation of disease duration with cognitive impairment is controversial. The study of pathologically confirmed cases of MSA showed that the mean time interval from disease onset to clinically significant cognitive impairment is seven years. Other studies showed cases with cognitive decline preceding motor impairment or early cognitive decline following motor impairment [3]. The prevalence of cognitive impairment was almost 50% in MSA patients surviving more than 8 years [10]. How cognitive impairment influents the disease duration is unclear.

Anxiety and depression are both frequently found in MSA patients [5,11,12]. For different motor subtypes, one study showed that depression and anxiety were both prominent in MSA-P, while there was only a higher level of anxiety in MSA-C. Anxiety and depression were related to cognitive impairment, such as executive regulation, abstraction, and learning [13]. In addition, orthostatic hypotension (OH) may be associated with cognitive impairment in MSA, although the results were not consistent [9,10,14,15]. The association of cognitive impairment with the severity of motor symptoms in MSA is not well understood. A large cohort MSA study showed that patients with cognitive impairment were older with a more severe motor disability [10]. Kim et al. showed that motor disability was more severe in MSA with dementia [16]. However, some patients showed rapid progress and had shorter survival times with severe motor symptoms and cardiovascular dysfunctions without significant cognitive impairment.

Therefore, we evaluated the cognitive function in detail in MSA without dementia patients using the Mini-Mental State Examination (MMSE) and the Chinese version of the Montreal Cognitive Assessment (MoCA) tests. Furthermore, clinical features between MSA patients with MCI and without MCI were assessed. Cognitive domains were compared between MSA-MCI and MSA without MCI patients, MSA-P, and MSA-C phenotypes separately using the MoCA test.

## 2. Materials and Methods

### 2.1. Participants

This was a cross-sectional descriptive study. All the patients who were first clinically diagnosed with possible and probable MSA in the department of Neurology, Tianjin Huanhu hospital and the second hospital of Tianjin Medical University between January 2016 and January 2022 were enrolled in this study. Patients were clinically diagnosed with possible and probable MSA with predominant parkinsonism (MSA-P) or MSA with predominant cerebellar ataxia (MSA-C) by movement disorder specialists according to the revised Gilman criteria [2]. A brain MRI scan was performed in all MSA patients. Patients who were illiterate or had diseases that may impair cognitive evaluation, such as stroke, hydrocephalus, brain tumor, epilepsy, or suspected MCI due to Alzheimer’s disease or others, were excluded. Patients were excluded if they had a family history of ataxia or parkinsonism, an established acquired etiology of ataxia, were unable to communicate, or refused to participate in the study. The Medical Ethics Committee of Tianjin Huanhu Hospital approved this study and informed consent was obtained from all participants. Among the 211 consecutive MSA patients, 11 patients were defined as having MSA with dementia and excluded from the present study. Finally, 102 patients with MSA-MCI and 98 patients with MSA-NC were recruited for the final analysis.

### 2.2. Clinical Assessment

All the clinical data, such as age, gender, educational level, age of onset, disease duration, motor symptoms, nonmotor symptoms, and auxiliary examination results, were collected. Nonmotor symptoms included urinary symptoms and residual urine volume, constipation, hypohidrosis, drooling, sleep disorders, as well as mood and behavioral problems. Motor severity was assessed using the modified Hoehn and Yahr (H&Y) stage from the Movement Disorder Society Unified Parkinson’s Disease Rating Scale (MDS-UPDRS) [17]. Global disability was assessed using Part IV (global disability scale) from the Unified Multiple System Atrophy Rating Scale (UMSARS- IV) [18]. Clinical possible rapid eye movement sleep behavior disorder (cpRBD) was screened using the special RBD questionnaire [19]. Anxiety/depression were assessed using the Hamilton Anxiety Scale (HAMA) and Hamilton Depression Scale-17 (HADM-17) [20]. Constipation was defined according to item 5 in the Scales for Outcomes in Parkinson’s Disease-Autonomic (SCOPA-AUT) [21]. Blood pressure (BP) measurements were taken at 1, 3, and 5 min in an upright position and compared with the last measurement in the supine position (baseline) or head-up tilt test. OH was defined based on a reduction in systolic BP of at least 20 mmHg or diastolic BP of 10 mmHg within 3 min of standing [22].

### 2.3. Cognitive Assessment

Global cognitive function was assessed according to the Chinese version of MMSE and MoCA, and each domain was evaluated according to MoCA sub-scores. Dementia patients were excluded from the final analysis. MSA-Dementia diagnosis was based on their MMSE score and Clinical Dementia Rating [23]. MoCA is used to assess nine cognitive domains. The cut-off scores for MCI detection referred to the study in China in Qihao Guo [24]. For the patients who did not finish the MoCA test or were missing in medical records, the cut-off MMSE scores for MCI detection are 26 (primary school education) or 27 (junior and above).

### 2.4. Statistical Analysis

SPSS version 20.0 (IBM Corporation, Armonk, NY, USA) was used for statistical analysis. Continuous data are presented as the mean ± standard deviation (SD) or median (interquartile range, IQR). Categorical data are presented as frequencies and percentages. A comparison of means between MSA-MCI and MSA-NC patients was performed using the independent *t*-test or the Mann–Whitney U test, depending on whether the data were normally distributed. The chi-square was used for comparing proportions. To explore the factors associated with MCI in MSA patients, a binary logistic regression was used. Multivariate analyses were performed using logistic regression in MSA patients between patients with MCI and NC as the dependent variable. *p*-values < 0.05 were considered statistically significant.

## 3. Results

### 3.1. Demographic Characteristics

The demographic characteristics of patients with MSA are shown in Table 1. From the cohort, 150 (75%) subjects had a diagnosis of probable MSA and 50 (25%) patients had possible MSA. A total of 102 (51%) subjects with MSA-MCI were identified. MSA-MCI patients were older (62.2 ± 8.7 vs. 59.1 ± 9.0, *p* = 0.015), had a later age of onset (59.1 ± 8.4 vs. 56.3 ± 9.0, *p* = 0.024), and more frequent comorbidity of hypertension (*p* = 0.022) than MSA-NC subjects. Significant differences were not found in gender, disease duration, educational level, smoking history, drinking history, or comorbidities of diabetes mellitus and coronary heart disease.

### 3.2. Clinical Characteristics in MSA Patients

MSA-MCI patients had a higher frequency of motor symptom onset and MSA-P phenotype compared with the MSA-NC subjects (*p* < 0.05). The positive rate of OH in MSA-MCI patients was lower than in MSA-NC subjects (*p* = 0.027). The prevalence of depression/anxiety in MSA-MCI patients was higher than in MSA-NC subjects (*p* = 0.004). A significant difference was not observed in the positive rate of Babinski sign, tremor, and postural instability between MSA-MCI and MSA-NC patients. The other nonmotor characteristics such as urinary incontinence, voiding difficulties, constipation, REM sleep behavior disorder, drooling, and abnormal sweating, were comparable between MSA-MCI and MSA-NC patients. (Table 2)

The factors associated with MCI in MSA patients were further investigated (Table 3). From the above results, age, age at onset, hypertension history, onset symptoms, and clinical phenotypes, presented with OH and depression/anxiety, there were significant differences between MSA-MCI and MSA-NC groups. We selected these variables to complete further logistic regression analyses. Based on univariate logistic regression analyses, older age (OR = 1.040; 95% CI: 1.007–1.075; *p* = 0.016), hypertension (OR = 2.238; 95% CI: 1.132–4.425; *p* = 0.021), motor onset (Ref: autonomic onset) (OR = 2.933; 95% CI: 1.392–6.179; *p* = 0.005), MSA-P phenotype (OR = 1.758; 95% CI: 1.004–3.077; *p* = 0.048), OH (OR = 0.524; 95% CI: 0.295–0.931; *p* = 0.027), and depression/anxiety (OR = 2.399; 95% CI: 1.318–4.365; *p* = 0.004) were associated with MCI in MSA.

After adjusting for potential confounding factors including gender, standing systolic BP (SBP), and the above six factors, motor onset (OR = 3.483; 95% CI: 1.200–10.110; *p* = 0.022) remained an independent factor associated with MSA-MCI (Table 3) as demonstrated in multivariate logistic regression analysis.

### 3.3. Characteristics of Cognitive Impairment in MSA-MCI Patients

All 158 MSA patients had detailed MoCA test recordings including 80 MSA-MCI and 78 MSA-NC subjects. The scores in each cognitive domain among MSA-MCI and MSA-NC subjects are shown in Table 4. Significant differences were observed in visuospatial, executive, and attention functions in the MoCA between MSA-MCI and MSA-NC patients indicating these were the predominantly affected cognitive domains in MSA-MCI patients.

### 3.4. Cognitive Impairment in MSA-P and MSA-C Patients

The scores in each cognitive domain among MSA-P and MSA-C subjects are shown in Table 5. Significant differences were observed in total scores (19.5 ± 4.2 vs. 20.9 ± 3.9, *p* = 0.031) and memory (1.3 ± 1.1 vs. 1.9 ± 1.3, *p* = 0.004) in the MoCA between MSA-P and MSA-C patients.

## 4. Discussion

The results demonstrated that approximately 50% of MSA patients had MCI during clinical evaluation. Compared with MSA-NC patients, MSA-MCI subjects were older and had a later age of onset. Patients with MSA-MCI had a higher frequency of motor symptom onset and MSA-P phenotype compared with the MSA-NC subjects, while disease duration was not different between the two groups. Visuospatial, executive, and attention functions were the primarily affected cognitive domains in MSA-MCI patients. Motor onset was independently associated with MCI in MSA patients.

In the present study, only 5.2% of MSA patients were diagnosed with MSA-dementia and 51% of MSA patients had MCI in a large MSA cohort with a mean duration of two years from onset. The median MMSE total score for the study cohort was 26 points, an indicator of relatively preserved global cognitive function, which is in accordance with previous studies on cognition in MSA [25,26,27]. The MCI prevalence ranged from 32 to 47% in MSA patients but in dementia it was rare. Severe dementia has only been reported in 12% of MSA cases in a study applying the Movement Disorder Society Parkinson’s disease dementia criteria [28].

Executive, attention, and visuospatial functions were the most impaired cognitive domains in the present study which highly correlates with the domain of cognitive impairment in MSA described in the previous studies. Frontal-executive dysfunction was the most prominent cognitive disturbance in MSA, especially in MSA-P, affecting up to 49% of patients. Memory and visuospatial function impairment were also common in MSA [3]. In recent studies, a predominant impairment of executive functions and verbal memory in MSA patients was also reported [26,29]. As with most of the previous studies, the present study also suggests a subcortical pattern of cognitive dysfunction in MSA. Pathological studies showed widespread subcortical degenerative changes in MSA and the substantia nigra and putamen were mostly affected. The disruption of subcortico–cortical pathways is likely to mediate some of the cognitive disorders in MSA [6,26,29].

In the present study, patients with MSA-P presented a higher prevalence of MCI (58%) than the MSA-C (44%) phenotype. More severe and widespread cognitive dysfunction was observed in MSA-P than in MSA-C patients, which was consistent with previous studies [7,8]. The present study showed that a more prominent impairment of memory was observed in MSA-P than in MSA-C patients. In general, cognitive profile consensus between MSA-C and the MSA-P motor subtypes is lacking due to conflicting results [3]. The impaired spontaneous immediate verbal recall is a main feature of MSA-P [5,14], which was similar to the pattern of Parkinson’s disease. However, reports exist on a more prominent impairment of memory, executive functions, and attention deficit in MSA-C [9,30,31]. Fabian Maass et al. [32] showed that MSA-C patients presented poor performance in language items compared with MSA-P patients. Kawahara et al. [33], using computerized touch panel screening tests, showed a significant decline in beating the devil’s game in MSA patients, and a significant extension of the flipping cards game only in MSA-C patients. However, in standard test batteries only, MSA-C patients showed a cognitive decline. The different cognitive screen tests may influence the results. More consistent and special screening tests for MSA are needed in the future.

The present study results also showed that 59.2% of MSA patients had OH, and the positive rate of OH in MSA-MCI subjects was lower than in MSA-NC subjects. However, after adjustment for age, gender, and supine and standing SBP, OH was not an independent risk or protective factor for cognitive decline. The association of OH and cognition in MSA has been addressed in some studies with controversial findings [9,10,26,34,35]. These results suggested that the influence of cardiovascular autonomic failure on cognition is potentially acute and reversible. The lack of a long-term prospective study with a large MSA cohort is the main problem. A prospective study, in which the eventual time-dependent cumulative OH effects on cognition are investigated, is necessary.

In this study, symptoms of depression/anxiety were present in 35.8% of MSA patients. These results are consistent with previous findings, which reported a prevalence of depression ranging from 20 to 80% [5,6,11,36], and for anxiety up to 40% [11,12]. Depression/anxiety was observed in 45.1% of MSA-MCI patients, which was higher than in MSA-NC patients. However, after adjusting for other confounding factors, the difference was non-significant. These results indicated that cognitive impairment was influenced by mood disturbances and not attributed to depressive or anxiety disorders. For some NSA patients, symptoms of depression or anxiety were present several years before motor symptoms. A prospective study is needed to clarify the time-dependent mood effects on cognition.

In the present study, the onset of motor symptoms was most common (55.5%), followed by the simultaneous onset of motor and autonomic symptoms (23.5%), and an initial autonomic symptom (21.0%), which was similar to a previous study including a large sample from the Mayo clinic with initial motor symptoms (61%), followed by autonomic onset (28%), and combined motor and autonomic symptoms (11%). The study results also showed the initial onset of either motor or autonomic symptoms did not influence the length of survival [37]. Studies showed that cognitive impairment or dementia was associated with motor severity in MSA, even in the early stages [10,14,16]. How the initial symptoms influence cognitive decline is unclear. However, in the present study, motor onset was an independent factor associated with MCI in MSA patients. Cognitive impairment in MSA is probably due to the involvement of prefrontal areas [7,38] causing striato-frontal dysfunction [9] or severe α-Syn pathology in the hippocampus [6,39,40]. Over the past few decades, research has established that the cerebellum is involved in executive memory and visuospatial and language functions, thus, indicating the cerebellum plays an important role in cognitive functions [41,42]. Motor onset indicates the early involvement of brainstem, cerebellum, or striatum, which may cause striato-frontal dysfunction or cerebellar cognitive dysfunction.

The present study had several limitations. First, the cut-off value of the MoCA to define MCI was based on studies of Parkinson’s disease (PD). Future studies are needed with detailed neuropsychological tests to define the optimal cut-off value for the detection of MCI in MSA patients. Furthermore, patients who were all clinically diagnosed lacked a neuropathological diagnosis, which was possibly leading to the misclassification of patient subgroups. Second, the MSA patients were all enrolled from the department of neurology with obvious symptoms, and due to the retrospective nature of the study, a standardized study protocol was not available; therefore, a reporting bias cannot be excluded. The prevalence number of MCI and other variables may be overestimated. Third, because of a lack of some important neuropsychological assessments such as memory with the Rey Auditory Verbal Learning Test, immediate recall and delayed recall, and a test for attention and executive function, more detailed cognitive analyses cannot be performed. More detailed neuropsychological assessments are needed in the following prospective study.

## 5. Conclusions

The results of the current study showed that MSA with MCI was observed in 51% of patients, with predominant impairment in visuospatial, executive, and attention functions. A more prominent memory impairment was observed in MSA-P than in MSA-C patients. Motor onset was independently associated with MCI in MSA patients. Future follow-up studies are warranted to identify more factors that predict the transition from MSA-MCI to MSA-dementia.

## Figures and Tables

**Table 1 brainsci-13-00582-t001:** Demographic characteristics in patients with MSA.

	Overall	MSA-MCI (N = 102)	MSA-NC (N = 98)	χ^2^/t/Z	*p*
Gender, Male *n (*%)	108 (54.0)	52 (48.1)	56 (51.9)	0.764	0.382
Female *n (*%)	92 (46.0)	50 (54.3)	42 (45.7)		
Age, Y	60.7 (9.0)	62.2 (8.7)	59.1 (9.0)	−2.459	0.015 *
Disease duration, M	24 (18–48)	30 (18–60)	24 (20–36)		0.456
Age at onset, Y	57.8 (8.8)	59.1 (8.4)	56.3 (9.0)	−2.281	0.024 *
Educational level, Y	9 (7,12)	8.5 (7.5,12)	9 (7,12)		0.679
MMSE	26 (24,28)	24 (22,25)	28 (27,29)		0.000 **
Diagnosis *n* (%)				3.228	0.072
probable MSA	150 (75.0)	71 (47.3)	79 (52.7)		
Possible MSA	50 (25.0)	31 (62.0)	19 (38.0)		
Smoking history *n* (%)	32 (20.0)	16 (19.5)	16 (20.5)	0.306	0.858
Drinking history *n* (%)	32 (20.0)	13 (16.0)	19 (24.4)	1.919	0.383
Comorbidity *n* (%)					
Hypertension	46 (28.6)	30 (36.6)	16 (20.3)	5.259	0.022 *
Diabetes mellitus	23 (14.3)	14 (17.1)	9 (11.4)	1.061	0.588
Coronary heart disease	13 (8.1)	7 (8.5)	6 (7.6)	0.417	0.812

Abbreviations: MSA-MCI, multiple-system atrophy with mild cognitive impairment; MSA-NC, multiple-system atrophy with normal cognition; Y, years; M, months; MMSE, Mini-Mental State Examination; * *p* < 0.05; ** *p* < 0.01.

**Table 2 brainsci-13-00582-t002:** Clinical features in MSA patients.

	Overall	MSA-MCI	MSA-NC	χ^2^/t/Z	*p*
Onset symptoms *n* (%)				8.755	0.013 *
Autonomic onset	42 (21.0)	14 (13.7)	28 (28.6)		
Motor onset	111 (55.5)	66 (64.7)	45 (45.9)		
Combined	47 (23.5)	22 (21.6)	25 (25.5)		
Clinical phenotype *n* (%)				3.922	0.048 *
MSA-C	100 (50)	44 (44.0)	56 (56.0)		
MSA-P	100 (50)	58 (58.0)	42 (42.0)		
Hoehn and Yahr	3.1 (0.7)	3.2 (0.7)	3.0 (0.6)	−1.801	0.073
UMSARS-IV	3.0 (0.7)	3.1 (0.7)	2.9 (0.7)	−1.835	0.068
Babinski sign *n* (%)	52 (39.1)	30 (36.1)	33 (41.8)	1.345	0.510
tremor *n* (%)	103 (51.5)	54 (52.9)	49 (59)	0.173	0.677
Postural instability *n* (%)	149 (74.5)	74 (72.5)	75 (76.5)	0.417	0.518
OH *n* (%)	119 (59.5)	53 (52.0)	66 (67.3)	4.910	0.027 *
Supine SBP	136.3 (20)	138.7 (21.4)	133.4 (18.2)	1.372	0.173
Supine DBP	84 (77,95)	86 (79,96)	83 (74,94)	1.464	0.404
Standing SBP	118.9 (24.1)	123.0 (24.2)	114.1 (23.5)	1.933	0.056
Standing DBP	78 (69,87)	80 (70,88)	77 (63,86)	1.573	0.148
Urinary incontinence *n* (%)	117 (58.5)	56 (54.9)	61 (62.2)	1.110	0.292
Voiding difficulties *n* (%)	54 (27)	26 (25.5)	28 (28.6)	0.241	0.624
Constipation *n* (%)	104 (62.7)	55 (63.2)	49 (62.0)	0.979	0.613
RBD *n* (%)	92 (46.0)	49 (48.0)	43 (43.8)	0.527	0.468
Drooling *n* (%)	30 (15)	18 (17.6)	12 (12.2)	0.873	0.350
Abnormal sweating *n* (%)	27 (13.5)	17 (16.5)	10 (10.2)	1.788	0.181
Depression/anxiety *n* (%)	71 (35.5)	46 (45.1)	25 (25.5)	8.375	0.004 **

Abbreviations: MSA-MCI, multiple-system atrophy with mild cognitive impairment; MSA-NC, multiple-system atrophy with normal cognition; MSA-C, MSA with predominant cerebellar ataxia; MSA-P, MSA with predominant parkinsonism; OH, orthostatic hypotension; SBP, systolic blood pressure; DBP, diastolic blood pressure; RBD, REM sleep behavior disorder; * *p* < 0.05; ** *p* < 0.01.

**Table 3 brainsci-13-00582-t003:** Logistic regression analyses for the factors associated with MSA-MCI.

Variable	Crude-OR	95% CI	Adjusted-OR	95% CI
Age, years	1.040 *	1.007–1.075	1.036	0.878–1.223
Age onset, years	1.038 *	1.005–1.073	0.994	0.842–1.175
Hypertension	2.238*	1.132–4.425	1.608	0.776–3.333
Motor onset (Ref: Autonomic onset)	2.933 **	1.392–6.179	2.582 *	1.181–5.645
Clinical phenotype, MSA-P (Ref: MSA-C)	1.758 *	1.004–3.077	1.285	0.690–2.390
OH	0.524 *	0.295–0.931	0.559	0.302–1.033
Depression/anxiety	2.399 **	1.318–4.365	1.878	0.976–3.612

Abbreviations: MSA-MCI, multiple-system atrophy with mild cognitive impairment; MSA-C, MSA with predominant cerebellar ataxia; MSA-P, MSA with predominant parkinsonism; OH, orthostatic hypotension; * *p* < 0.05; ** *p* < 0.01.

**Table 4 brainsci-13-00582-t004:** Cognitive profiles in patients with MSA-MCI on the MoCA.

	MSA-MCI (N = 80)	MSA-NC (N = 78)	t	*p*
Visual perception and Executive function (5)	1.8 (1.3)	3.0 (1.3)	2.142	0.047 *
Naming (3)	2.8 (0.5)	2.6 (0.8)	0.471	0.925
Attention (3)	1.9 (1.0)	2.9 (0.4)	3.197	0.040 *
Calculation (3)	2.2 (1.1)	2.9 (0.4)	2.101	0.238
Language (3)	1.4 (0.9)	1.9 (1.0)	1.157	0.238
Conceptual thinking (2)	1.2 (0.9)	1.5 (0.8)	0.828	0.466
Memory (5)	1.3 (1.3)	1.8 (1.5)	0.769	0.506
Orientation (6)	5.2 (1.0)	5.9 (0.4)	2.326	0.101
MOCA total (30)	17.8 (3.3)	22.5 (3.5)	2.996	0.009 **

Abbreviations: MSA-MCI, multiple-system atrophy with mild cognitive impairment; MSA-NC, multiple-system atrophy with normal cognition; MoCA, Montreal Cognitive Assessment; * *p* < 0.05; ** *p* < 0.01. The numbers under the columns MSA-NC and MSA-MCI: Mean (Standard deviation).

**Table 5 brainsci-13-00582-t005:** Cognitive profiles in MSA-P and MSA-C patients on the MoCA.

	MSA-P (N = 81)	MSA-C (N = 77)	t	*p*
Visual perception and Executive function (5)	2.2 (1.3)	2.4(1.2)	0.832	0.407
Naming (3)	2.6 (0.7)	2.7 (0.6)	1.438	0.152
Attention (3)	2.3 (0.9)	2.4 (0.9)	0.968	0.334
Calculation (3)	2.6 (0.7)	2.8 (0.4)	1.950	0.053
Language (3)	1.6 (1.0)	1.8 (1.0)	1.100	0.272
Conceptual thinking (2)	1.2 (0.8)	1.3 (0.8)	0.820	0.414
Memory (5)	1.3 (1.1)	1.9 (1.3)	2.937	0.004 **
Orientation (6)	5.6 (0.7)	5.6 (0.6)	0.938	0.350
MOCA total (30)	19.5 (4.2)	20.9 (3.9)	2.180	0.031 *

Abbreviations: MSA-C, multiple-system atrophy with predominant cerebellar ataxia; MSA-P, multiple-system atrophy with predominant parkinsonism; MoCA, Montreal Cognitive Assessment; * *p* < 0.05; ** *p* < 0.01. The numbers under the columns MSA-P and MSA-C: Mean (Standard deviation).

## Data Availability

The datasets used and analyzed during the current study are available from the corresponding author on reasonable request.

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
