# Peer review of "Characteristics of Mild Cognitive Impairment and Associated Factors in MSA Patients"

_brainsci, 2023, doi:10.3390/brainsci13040582_

Round 1

Reviewer 1 Report

Congratulation for the unique finding of MCI characteristics among the MSA population. I have only suggested that you should use more neuropsychological assessments like memory with the Rey Auditory Verbal Learning Test (RAVLT)—immediate recall (RAVLT IR) and delayed recall (RAVLT DR), test for attention, executive function etc. As I understand, this was impossible because of the study's retrospective nature, and you need to mention this in the limitation.

I also mentioned that there was a lack of assessment of activities of daily living.

Reviewer 2 Report

This is an interesting study assessing cognitive functions in patients with MSA. The originality of this study is related to the comparison between patients with cognitive dysfunction and those who according to neuropsychological tests have normal cognition. The obtained results confirm known facts that age, arteriosclerotic co-morbidities (hypertension) are related to cognitive deficits. The authors have shown that clinical phenotype MSA-P is more often related to deficit of cognition compared to MSA-C. It has to be noted, however, that this observation was already published long time ago (Kawai et al 2008).

In my opinion one very important analysis of the results is missing – comparison of cognitive profiles between MSA-P and MSA-C. Are there differences in the results of the assessments of psychological domains (e.g. are executive functions more often affected in MSA-P compared to MSA-C, the same question concerns all other domains)? As such studies are rather rare in the literature and the results are controversial  (Koga et al 2017, Kawahara et al 2015), this type of the analysis of the obtained results would make this study much better.

Reviewer 3 Report

Dear authors,

Thank you for your valuable research.

The following items are suggested to improve the scientific level of the manuscript.

Please:

1- In the introduction, briefly talk about the factors affecting cognition in MSA patients. So that the need to check the variables is clear for the reader.

2- In the method section, the type of study should be mentioned in the first line (cohort or cross-sectional descriptive).

3- Strengthen the discussion section by addressing research hypotheses and mentioning more references.

Best regards,

Reviewer 4 Report

This is an interesting study examining characteristics of mild cognitive impairment and associated factors in multiple system atrophy patients. The study is promising but I have several major concerns regarding the study:

1. First, the literature review of this paper is too short and there is no strong justification on the purpose of the current study. There is a need for the authors to expand the introduction, present several research gaps, and properly hypothesize their study.

2. There is a need for the authors to elaborate further the inclusion and exclusion criteria of the study. They also need to clarify how participants were recruited.

3. There is a need to clarify further how Bonferroni correction was applied. Is it to all the analyses?

4. The prevalence number discussed in the current study is biased as the study used convenience sampling. This is something that should be discussed in the discussion section.

5. There is a need to clarify how the specific variables in Table 3 were chosen.

Round 2

Reviewer 2 Report

In my opinion the manuscript, the authors had made the suitable corrections.

Author Response

thanks lot. I will aply to english edit to check the English language.

Reviewer 4 Report

I appreciate the authors' efforts in revising the manuscript. I still have some concerns regarding the manuscript and I hope that the authors can improve  the manuscript further before it is published. I understand that a few sentences have been added in the Introduction. But many of the newly added sentences are incomplete and should be expanded further. For example, the authors mentioned that The association of cognitive impairment with severity of motor symptoms in MSA is not well understood. But the next sentence doesn't flow with this argument. I think a further revision of the introduction is necessary.

Also, the abstract can be improved further by having a final sentence highlighting the overall contribution of the study.
